# Natural Killer Cells, as the Rising Point in Tissues, Are Forgotten in the Kidney

**DOI:** 10.3390/biom13050748

**Published:** 2023-04-26

**Authors:** Ke Ma, Zi-Run Zheng, Yu Meng

**Affiliations:** 1Department of Nephrology, The First Affiliated Hospital of Jinan University, Guangzhou 510000, China; 2Department of Nephrology, The Fifth Affiliated Hospital of Jinan University, Heyuan 570000, China

**Keywords:** natural killer cells, kidney, tissues

## Abstract

Natural killer (NK) cells are members of a rapidly expanding family of innate lymphoid cells (ILCs). NK cells play roles in the spleen, periphery, and in many tissues, such as the liver, uterine, lung, adipose, and so on. While the immunological functions of NK cells are well established in these organs, comparatively little is known about NK cells in the kidney. Our understanding of NK cells is rapidly rising, with more and more studies highlighting the functional significance of NK cells in different types of kidney diseases. Recent progress has been made in translating these findings to clinical diseases that occur in the kidney, with indications of subset-specific roles of NK cells in the kidney. For the development of targeted therapeutics to delay kidney disease progression, a better understanding of the NK cell with respect to the mechanisms of kidney diseases is necessary. In order to promote the targeted treatment ability of NK cells in clinical diseases, in this paper we demonstrate the roles that NK cells play in different organs, especially the functions of NK cells in the kidney.

## 1. Introduction

NK cells are innate lymphocyte cells (ILCs), not only being critical in anti-viral and tumor defense but also playing significant roles in regulating homeostasis in tissues such as the liver, adipose, and kidney [1,2,3,4,5,6,7,8,9,10]. Unlike T and B cells that recognize antigens through clonally distributed, somatically rearranged receptors, NK cells recognize their targets by integrating signals from multiple germline-encoded activating and inhibitory receptors that recognize major histocompatibility complex class I (MHC class I) molecules [3,4,7,10,11,12,13,14,15,16,17,18]. In acute and persistent viral infections, NK cells are necessary to control viruses such as Herpesviruses, while in cancer, NK cells promote the direct dissolution of various tumor types and inspect different sites to prevent metastasis [8,9,10,16,17,18,19,20,21]. These indicate that NK cells play a dynamic role in immune-mediated protection and homeostasis.

It has been clear that tissue localization is a critical determinant of the function and in vivo role of lymphocytes, which indicates that tissue diseases are principally regarded as being associated with the abnormal activation of T cells, B cells, macrophages, NK cells, etc. [14,17]. With the deepening of research, NK cells have been found to play an important role in clinical diseases by linking innate and adaptive immunity, for example in autoimmune hepatitis, type 1 diabetes mellitus (T1DM), and obesity [1,2,22,23,24,25,26,27]. NK cells can be recognized with the CD56^+^CD3^−^ phenotype [28,29,30]. Based on the expression level of CD56, two subsets of NK cells, CD56^bright^ and CD56^dim^, have been introduced [31]. CD56^bright^, with cytokine production and imperfect cytotoxic activity, only comprises 10% of peripheral blood NK cells (pNKs) [15,16,32,33], whereas CD56^dim^, with the totally mature feature of short telomere lengths, has high cytotoxic activity and represents approximately 90% of pNKs [31,34]. NK cells defend against infections and tumors by the integration of activating and inhibitory signals from NK cell surface receptors [35,36,37]. For one thing, NK cells detect malignancy cells and infected cells, then rapidly respond by secreting cytotoxic granules or death receptor ligands [31]. For another, they also exert immunoregulatory functions in innate and adaptive immune responses by producing many types of cytokines and chemokines such as interferon-γ (IFN-γ), tumor necrosis factor-alpha (TNF-α), GM-CSF, interleukin (IL)-10, and IL-13 [22,31,38,39,40,41]. They also exhibit immunologic memory, which challenges the conventional distinctions between innate and adaptive immunity [42,43,44,45,46]. However, the hyperactivation or dysfunction of NK cells is associated with the pathogenesis of various inflammatory and autoimmune diseases. Thus, NK cells have both protective and pathogenic functions in clinical diseases depending on the NK cell subset, microenvironment, disease type and developmental stage [47]. Additionally, the role of tissue localization in human NK cell development and function, and how circulating NK cells relate to those in different sites, are not well understood. Understanding the distribution, function, and development of human NK cell subsets across multiple tissue sites, ages, and individuals will provide essential insights into their role in immune responses and surveillance.

Although numerous studies have shown different roles of pNKs and trNKs at different tissue sites, only a few studies have pointed to an important role of NK cells in kidney disease, especially with respect to trNKs. In addition, it is still difficult for us to comprehend the role of NK cells in kidneys under healthy or diseased conditions. Here, we first summarize different tissue-specific patterns of NK cell subset distribution and their function. Secondly, we mainly discuss the function of NK cells in the kidney for a better understanding of NK cell within the mechanisms of kidney diseases, contributing to halting kidney disease progression and trying to provide new ideas for the targeted immunocyte therapy of kidney diseases [48].

## 2. Natural Killers Cells

NK cells were the first identified subtype of ILCs, responding to virally infected and/or transformed cells with a variety of effector functions, chiefly cell killing and the production of proinflammatory cytokines [3,4,49,50]. All ILC family members (such as ILC1s, ILC2s, and ILC3s) originate from the same common lymphoid progenitor cells as B cells and T cells [49,50]. Among these, NK cells belong to ILC1 and are essential in killing infected or transformed cells by mediating cytotoxicity and producing significant quantities of inflammatory cytokines, including IFN-γ and TNF-α [16,49,51,52,53,54,55,56,57]. Conventional NK cells (cNKs) are referred to as those NK cells obtained from the spleen or bone marrow (BM) in mice, and human peripheral blood mononuclear cells (PBMCs). Recent studies have begun providing insights into the less studied trNKs [57,58,59,60]. The dichotomy between the cNKs and trNKs is recent famous research.

Without the expression of CD56 in murine NK cells, they are always distinguished as NK1.1, NKp46, and/or CD49b (DX5), and maturity is recognized according to the expression of CD11b and CD27 [9,12]. Fortunately, most activating and inhibitory receptors are conserved between humans and mice, such as NKp46 and NKG2D [14]. CD49a is also used as a marker for trNKs in mice. However, trNKs were classified as ILC1s in most studies due to the lack of expression of the transcription factor Eomes in the mouse population [10]. Despite these differences, parabiosis mouse models have shown that 80–90% of lung NK cells are cNKs, with the remaining 10–20% presumed to be trNKs, in agreement with the human studies described above. Further studies are needed to ascertain the functional role of this potential trNK cell population [11,61].

Based on the expression of CD16 (FCRγIII), human cNK cells can be further divided into two subsets, CD56^bright^CD16^−^ and CD56^dim^CD16^+^. The distribution of these subsets among cNKs and trNKs differs significantly [3,5,6]. In mice, both cNKs and trILC1s are defined as CD45^+^NK1.1^+^NKp46^+^ lymphocytes, with cNKs being CD49a^−^CD49b^+^ and trILC1s CD49a^+^CD49b^−^. While cNKs rely on the transcription factors Eomes and Nfil3 for their development, trILC1s rather depend on T-bet, Hobit, and the aryl hydrocarbon receptor (AhR) [15,18,33,62,63]. Furthermore, cNKs express higher amounts of markers associated with maturation (CD11b, CD43, KLRG-1) compared to trILC1s, which rather feature the cytotoxic molecule TNF-related apoptosis-inducing ligand and markers for tissue residency (e.g., CD69) [64]. Additionally, the surfaces of NK cells contain a large number of receptors and receptor ligands.

The main functions of NK cells in various tissues have not been fully understood. Therefore, this review focuses on the role of trNKs in physiological and disease conditions. Recently, people are increasingly interested in characterizing and studying NK cells in tissues. NK cells can play a role not only in various tissues and organs but also in various diseases. Therefore, it is necessary for rapid research progress mainly including NK cells in the uterus, liver, lung, fat, and kidney. Here, we summarize the classic circulating NK cells and the resident NK cells of major organs such as the liver, kidney, lung, and uterus (Figure 1). In addition, we include the commonly activated receptors, inhibitory receptors, and cytokine receptors on the surfaces of NK cells in this paper (Table 1).

## 3. The Role of NK Cells in Different Tissues

The studies about the role NK cells play in different organs and diseases cause more and more research relating to the function of NK cells. Here, we demonstrate the role of NK cells in the uterus, liver, lung, adipose, and kidney (Table 2 and Figure 2).

### 3.1. NK Cells in the Uterus

Studies have demonstrated that adaptive lymphocytes locally differentiate in peripheral tissues with the appearance of tissue-resident memory cells [78,79]. Along the same lines, human NK cells can differentiate from CD56^bright^ to more differentiated CD56^dim^ NK cells in secondary lymphoid tissue and peripheral blood [80]. After that, CD56^dim^NK cells undergo a further maturation, which is marked by the loss of NKG2A and the acquisition of killer-cell-immunoglobulin-like receptors (KIRs) and CD57, reducing proliferative capacity and increasing natural cytotoxicity [81,82]. Outside of circulation, NK cells are enriched in the uterus [83]. A few studies have suggested critical roles for uterine NK cells (uNKs) in regulating trophoblast invasion and promoting arterial integrity, decidualization, EVT invasion, spiral arterial remodeling, early placental formation, and fetal growth [67,69,84]. Recently, studies have shown that the resident NK cells of the uterus in pregnancy and non-pregnancy also have their own specificity. In pregnancy conditions, uNK cells, also known as decidual NK cells (dNK), produce a variety of cytokines, growth factors, and vascular growth factors to ensure the smooth progress of the pregnancy [85,86,87,88,89,90,91,92,93]. The uNKs in a non-pregnant state are also called endometrial NK cells (eNK), which exhibit cytotoxicity (Figure 2) [86,89,90,91,92,93]. Benedikt Strunz et al. [67] showed that less differentiated KIR^−^CD39^−^ uNK cells constitute a more proinflammatory uNK cell subset capable of producing IFN-γ, CCL3, CCL4, and TNF, among which IFN-γ has been suggested to be important for spiral artery formation in murine reproduction. In contrast to this, functions exhibited by KIR^+^CD39^+^ uNK cells have been linked to immune tolerance and enhanced angiogenic capacity, with the production of galectin-1 and galectin-9, which might regulate fetomaternal tolerance and the vascularization of the placenta and decidua during early pregnancy. Emily M. Whettlock et al. [69] demonstrated that uNK can be divided into three subsets (uNK1, uNK2, uNK3), which may have different roles in pregnancy. UNK1s express higher levels of KIRs and LILRB1, indicating their role in communicating with EVTs. UNK2 and uNK3 produced more cytokines upon stimulation, indicating their role in immune defense. It has been found that uNK1 and uNK2 peak in the first trimester, while uNK3 matters a lot in the third trimester. All three subsets are most capable of degranulation and cytokine production during the secretory phase of the menstrual cycle and express KIR2D molecules, which enables them to interact with HLA-C expressed by placental extravillous trophoblast cells.

### 3.2. NK Cells in the Liver

The liver has been proposed to be an innate immune organ, containing a large number of innate immune cells [63,64,78]. In particular, NK cells are abundantly present within the liver, accounting for up to 30–50% of the total intrahepatic lymphocytes in humans [79]. Moreover, accumulating evidence has shown that hepatic NK cells display unique phenotypic and functional characteristics, differing from peripheral NK cells in terms of subset composition at a steady state [24,43,80].

Recent studies have revealed a unique NK cell subset, termed liver-resident NK (lrNK) cells, and have shown that mouse liver NK cells can be divided into two populations, lrNK cells and conventional NK (cNK) cells, based on the mutually exclusive expression of CD49a and CD49b (DX5). Among them, lrNK cells are characterized by a CD49a^+^CD49b^–^ phenotype [24]. LrNKs have been shown not only to be phenotypically different from their CD56^bright^ cNK brethren, but also transcriptionally distinct, and as with other tissues, these resident NK cells appear to be specifically suited to function in the liver environment and integral to the regulation of immune function in this organ [81,82]. Here, we discuss liver NK cells and their involvement in the pathogenesis of liver inflammation and diseases following dysregulation (Figure 2) [83].

Functionally, CXCR6^+^ lrNK cells produce less IFNγ, tumor necrosis factor (TNF), and macrophage inflammatory protein (MIP)-1β in response to stimulation than hepatic cNK cells. They also express less perforin and Granzyme B but show enhanced Granzyme K expression and are capable of degranulation [84]. Finally, a study investigating hepatic NK cells in liver transplants ascertained that, although CXCR6^+^ lrNK cells are long-lived and do not recirculate, they could still be replenished from circulation [84]. In addition to CXCR6^+^ CD56^bright^ CD16^-^ lrNK cells, a second liver-resident NK cell population has been described [67]. These cells are CD56^bright^ CD16-, lacking CXCR6 expression and instead being identified by the expression of CD49a, similar to mouse ILC1s [24,67]. CD49a^+^ lrNKs were present in approximately 40% of the tested individuals with a frequency of approximately 2.3% of hepatic CD56^bright^ cells. CD56^bright^ CD49a^+^ NK cells specifically localized to the liver parenchyma and, unlike CXCR6^+^ lrNKs, expressed T-bet but not Eomes. They also exhibited a pattern of KIR and NKG2C expression indicative of clonal-like expansion, although they did not express CD57, a marker expressed on differentiated cNK cells. Furthermore, CD56^bright^ CD49a^+^ lrNK cells had reduced capacity to degranulate compared to hepatic CD49a- NK cells but were potent producers of cytokines such as IFNγ, TNF, and GM-CSF [83]. The T-cell-immunoglobulin-and-mucin-domain-containing protein 3 (Tim-3) was significantly upregulated in both tumor-infiltrating lrNK and cNK cells and suppressed their cytokine secretion and cytotoxic activity, thereby enhancing hepatocellular carcinoma growth by blocking natural killer cell function [69,85].

### 3.3. NK Cells in the Lung

In a study associated with lung-resident NK cells, it identified adaptive-like NK cell expansions with tissue-resident traits in the lung and blood in approximately 20% of individuals, which were hyperresponsive to target cell stimulation, differing from adaptive-like CD16+ blood NK cells. Expanded NK cells in this particular NK cell subset will likely induce a different course of acute lung disease such as viral infections, which may be a future tool in the treatment of lung cancer [71]. The lungs comprise mucosae that are constantly exposed to environmental and autologous stimuli, and they are sites of a high incidence of primary and metastatic tumors [86,87]. Accordingly, a rapid and efficient immune response that prevents tumorigenesis and pathogen invasion without leading to excessive inflammation is needed to maintain pulmonary homeostasis. As a type of innate immune cell, natural killer (NK) cells are regarded as the host’s first line of defense against tumors and viral infection [87]. Moreover, the involvement of NK cells in various lung diseases, such as lung cancer, chronic obstructive pulmonary disease (COPD), and asthma, as well as infections, has been documented [71,88,89,90,91,92,93]. Relevant studies have shown that NK cells constitute 10% of the lymphocytes in the human lung, superior to other tissues such as the spleen, peritoneal exudates, and lymph nodes. Human lung NK cells are mostly circulating CD56^dim^CD16^+^ cells. By contrast, a minority of NK cells are tissue-resident, CD56^brigh^tCD16^-^NK cells, characterized by a high expression of CD49a, CD69, and CD103. Surprisingly, it is predicted that CD49a^+^CD103^+^trNK cells have an intraepithelial location in the human lung, although the exact localization of the different trNK cell subsets remains to be investigated. In the study of primary graft dysfunction (PGD) in human lung transplantation, CD16^+^ NK cells were increased in absolute and relative quantities for up to 90 days following severe PGD. As a marker of NK cell maturation, CD16 is an activating Fc receptor and initiates antibody-dependent cell-mediated cytotoxicity (ADCC) [73]. Nicole Marquardt et al. [94] showed that lung CD56^dim^ NK cells were markedly hyporesponsive to stimulation with K562 cells, a prototypic tumor target cell line lacking HLA class I expression, compared with CD56^dim^ NK cells in matched peripheral blood, indicating that the former was impaired both in terms of natural cytotoxicity and ADCC. NK cells in the lung have a better capacity to be hyper-functional, so they are regarded as the host’s first line of defense against various lung diseases.

### 3.4. NK Cells in Adipose Tissue

Adipose tissue comprises multiple types of immune cells, which connect health and disease strictly with adiposity and metabolism [65,66,68,70,73,94]. According to the data of many single-cell sequencing and biochemical analyses, we found that adipose tissue contains a variety of immune cells and cytokines, which play a huge role in metabolic-related diseases such as diabetes, obese glomerulonephritis, etc., through complex crosstalk [72,74].

Previous studies have shown that macrophages, neutrophils, eosinophils, innate lymphoid type 2 cells, dendritic cells, and mast cells in adipose tissue all participate in the disease mechanism. However, as time has gone by, NK cells have been found to play an important role in diseases related to obesity in adipose tissue. Wensveen et al. [75] discovered that NK cells link obesity-induced adipose stress to inflammation and insulin resistance by accumulating proinflammatory macrophages in adipose tissue. Byung-Cheol Lee et al. showed that a high-fat diet (HFD) increases NK cell numbers and the production of proinflammatory cytokines [76].

## 4. Natural Killer Cells in the Kidney

### 4.1. NK Cells in a Steady Kidney

Most of our knowledge regarding NK cells is derived from studies on peripheral blood and lymphoid organs. However, it is now established that NK cells also exist in non-lymphoid organs of the human body, including the kidney, with the expression of different phenotypes. Factually, NK cells constitute a large fraction of total lymphocytes in healthy human kidneys (~25% of lymphocytes) [95,96]. In our study, the single-cell suspension prepared from C57 mice kidneys was incubated with flow cytometry fluorescent antibody. After that, we obtained the main proportion and number of immune cells and NK cell subsets in the kidney through Flow Jo analysis. Our flow cytometry results showed that renal NK cells accounted for about 5% of renal lymphocytes, of which NK cells accounted for about 9–10% of lymphocytes in the kidney (Figure 3). Moreover, we also observed the distribution of NK cells through immunofluorescence from kidney samples. Most of the NK cells were mainly distributed on the glomerular basement membrane, and the rest were scattered around the renal capillaries and near the glomerulus. The distribution of classical circulating immune cells was similar to that of NK cells.

### 4.2. NK Cells in Acute Kidney Injury

Acute kidney injury (AKI) is a clinical condition characterized by the acute impairment of kidney function and is induced by different causes, including ischemia, sepsis, and toxic insults. In particular, ischemia–reperfusion injury (IRI) is one of the most frequent events leading to severe AKI [97]. A large number of studies have demonstrated that NK cells take part in IRI–AKI by different pathways (Figure 4). Recently, works have demonstrated that NK cells can injure tubular epithelial cells (TECs) and contribute to kidney ischemia–reperfusion injury (IRI) in the absence of both T and B cells, which means NK cells also play an important role in IRI. [98,99,100]. Victorino et al. [101] showed us that depleting NK cells can protect mice from kidney dysfunction during IRI, and tissue-resident NK cells promote AKI. Zhang et al. [100] proved that NK cells cause tubular epithelial cell (TEC) apoptosis and contribute to renal IRI after being activated by osteopontin. Other studies reported that NK cells induced initial kidney inflammation during IRI [102,103], and after 3 and 24 h of renal IRI, the increased infiltration of NK1.1^+^ and CD4^+^NK1.1^+^ cells was observed in kidney IRI [104]. CD49b^+^NKG2D^+^ cNK cells were shown to promote tubular injury induced by the ischemic insult, a finding that depended on the NK cell expression of the cytotoxic effector molecule perforin [99]. In addition, NK cells also can lead to TEC injury by recruiting neutrophils [102]. TEC and NK cell crosstalk occurs through different receptor–ligand pairs such as NKG2D ligands (NKG2D-L) and CD137–CD137L inducing cytotoxic activity and interferon (IFN)-γ production, and stimulates the secretion of chemokines attracting neutrophils in TECs [99,103]. TECs are killed by NK cells through the release of cytotoxic granules, while activated neutrophils are responsible for tissue damage due to reactive oxygen species (ROS) and lytic enzymes (Table 3) [105].

### 4.3. NK Cells in Adriamycin Nephropathy

Chronic kidney disease (CKD) has been considered to involve renal endothelial cell injury and microvascular dysfunction and fibrosis. Adriamycin (ADR) causes progressive proteinuria by inducing glomerular endothelial dysfunction. Zheng et al. proved that in the ADR model, the absence of functional B and T cells as well as residual NK cells and macrophages in severe combined immunodeficient (SCID) mice resulted in more severe disease than in immunocompetent mice by upregulating activating receptor NKG2D and its ligand RAE-1.

### 4.4. NK Cells in Renal Cell Carcinoma

Renal cell carcinoma (RCC) already represents the sixth most frequently diagnosed cancer in men and the 10th in women in the world [109,110,111,112]. The incidence rates of RCC are always increasing [109,110,111,112]. NK cells, as a part of the first defense against neoplastic growth, play an important role in inducing tumorous cell death via cytokine and chemokine release [113,114,115,116]. Recently, studies have demonstrated that NK cells are a part of renal cell carcinoma as a non-classical phenotype, constituting about 20% of immune cells. The survival of the renal cell tumor is closely correlated with the frequency of NK cells. It has been discovered that the detection of NK cells can help evaluate the degree of malignancy and may predict the prognosis of renal cell carcinoma. Nevertheless, there are RCCs with NK cell infiltration, meaning the tumor is still metastasizing elsewhere. It is predicted that the tumor can escape the surveillance of NK cells (Figure 5), and potential targets and interventions must further be explored to solve this. NK cell populations, according to differences in gene expression profiles, can be divided into three subsets: NK(GZMH), NK(EGR1), and NK(CAPG), and the latter two subsets are closely related to tumor metastasis [117]. Schleypen et al. [107] found that NK cells express various inhibitory receptors (IRs) such as CD94/NKG2A receptor complex in RCC. In addition, in the subsequent studies, Schleypen et al. [108] discovered that the markers on NK cells such as CD16 can be associated with their function in RCC.

## 5. Discussion and Future Perspectives

The existing evidence shows that NK cells play an important role in various tissues and organs, such as the uterus, liver, lung, fat, etc. In the uterus, NK cells play an indispensable role in remodeling endometrial arteries. In the liver, NK cells can also play a role in the occurrence and development of liver fibrosis. With the deepening of the research on NK cells, we unexpectedly found that research on NK cells in kidney diseases is still rare. We think this is very regrettable. By optimizing the processing method of kidney single-cell homogenate, we used flow cytometry to detect that the proportion of NK cells in lymphocytes in the kidney can reach about 10%. This is a very useful discovery. NK cells account for a large proportion of immune cells in the kidney, so they must have obvious immune effects. However, there are few studies on NK cells in kidney diseases. We found that NK cells promote disease progression in renal IRI, especially tissue-resident NK cells (trNK), by summarizing the existing literature. We also found that NK cells also play a certain role in clinical samples such as renal cell carcinoma and renal fibrosis. Therefore, we described NK cells in kidney diseases. This will provide a certain basis for clinical research on NK cells in order to change the disease’s progress in the future.

## Figures and Tables

**Figure 1 biomolecules-13-00748-f001:**
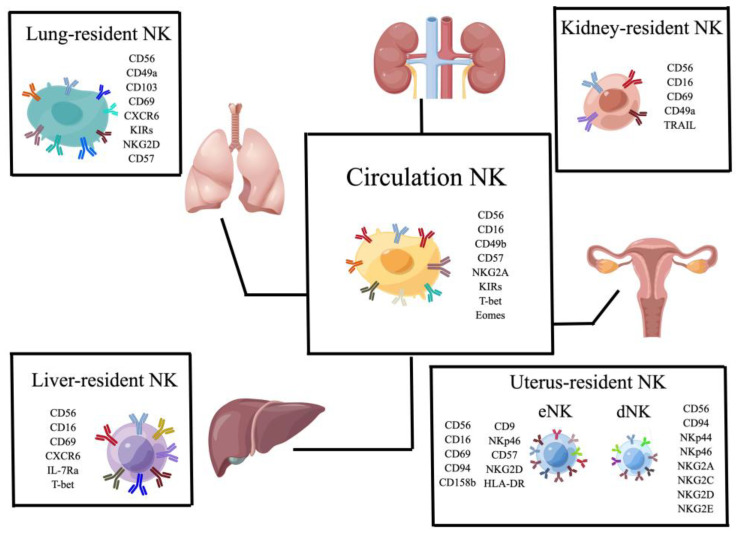
Classic circulating NK cells (cNK) and organ-specific tissue-resident NK cells in the kidney, liver, uterus, and lung.

**Figure 2 biomolecules-13-00748-f002:**
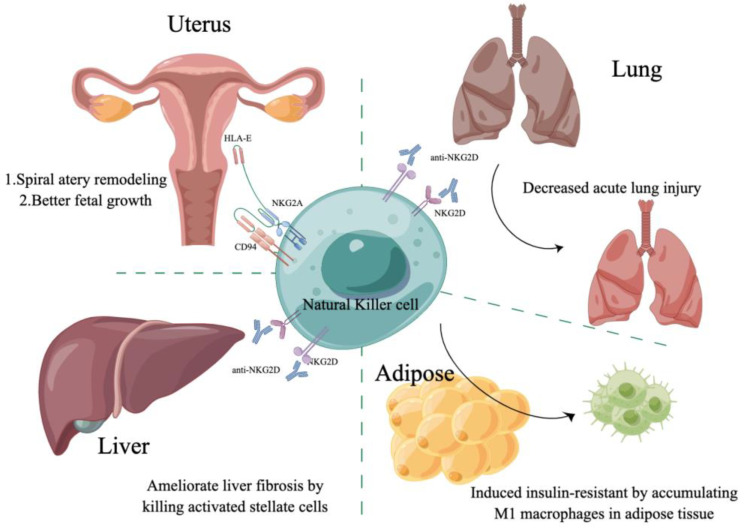
NK cells’ receptors combine with the ligands of different organs to participate in the occurrence and development of disease.

**Figure 3 biomolecules-13-00748-f003:**
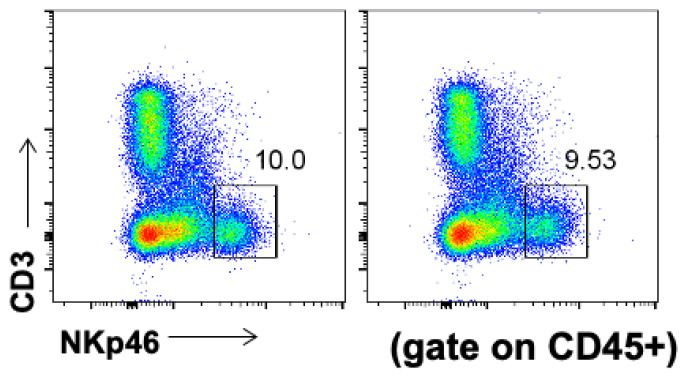
The NK cells accounted for lymphocytes in the kidney gated on CD45. Renal single-cell suspension was analyzed through a combination of mechanical and enzymatic digestion methods. Then, it was bound to flow antibodies (Fc blocker, Dead-7-aad, CD45-FITC, CD3-APC-CY7, NKp46-AF647) and finally analyzed by Flow Jo after examining FACS Calibur (BD Biosciences). We obtained about 10% NK cells in kidney lymphocytes.

**Figure 4 biomolecules-13-00748-f004:**
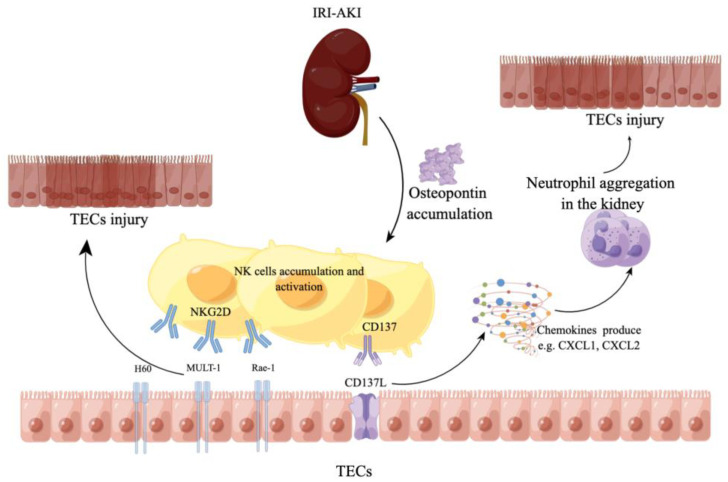
NK cells are involved in acute kidney disease by directly binding with the ligands of renal tubular epithelial cells or indirectly promoting the accumulation of neutrophils. (IRI–AKI: ischemia–reperfusion injury–acute kidney injury; TECs: tubular epithelial cells; CXCL1: chemokine (C-X-C motif) ligand 1; CXCL2: chemokine (C-X-C motif) ligand 2).

**Figure 5 biomolecules-13-00748-f005:**
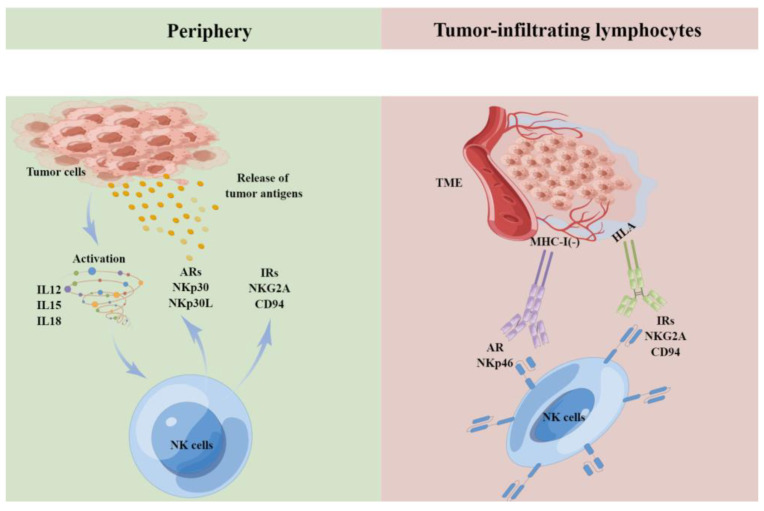
Activating and inhibitory signaling balance determines NK cell killing and cytokine. production. In the periphery and tumor-infiltrating-lymphocytes (TILs), NK cells express activating receptors (ARs) and inhibitory receptors (IRs) upon activation by target tumor cells or cytokines (IL12, IL15, IL18). NK cell activation will depend on the signal balance. HLA: human leukocyte antigen; MHC-I: MHC-class I negative.

**Table 1 biomolecules-13-00748-t001:** The main receptors in human NK cell surface.

Receptors Type	Surface Receptor in Human NK Cells
Activating Receptors	NKG2C NKG2D NKp30 NKp44 NKp46 NKp80 CD2 CD16 CD95L DNAM1(CD226) Activating KIR
Inhibitory Receptors	Inhibitory KIR TIGIT NKG2A NKG2B CD96 PD-1 KLRG-1 TIM-3
Cytokine Receptors	IL-1R IL-2R IL12R IL18R IL21R IFN-AR
Death Receptors Ligands	Fas-L TRAIL
Homing Receptors and Adhesion molecules	CCR2 CCR5 CCR7 CXCR1 CXCR3 CXCR4 CXCR6

**Table 2 biomolecules-13-00748-t002:** The function of NK cells in the uterine, lung, and liver.

Tissue	The Effect of NK Cells in Organs	Authors
Uterine	NK cells’ NKG2A pathway leads to a healthier pregnancy.	Shreeve et al. [65]
	Uterine NK (uNK) cells take part in remodeling of spiral arteries.	Wells et al. [66]
	uNK cell differentiation driven by interleukin-15 occurred to counter the endometrial regeneration.	Strunz et al. [67]
	significantly increased uNK level in endometrium of women with RM and RIF may point to an underlying disturbance of the immune milieu culminating in implantation and/or placentation failure.	Von Woon et al. [68]
	Dynamic Changes in Uterine NK Cell Subset Frequency and Function Over the Menstrual Cycle and Pregnancy	Whettlock et al. [69]
	Uterine NK cells underexpress KIR2DL1/S1 and LILRB1 in reproductive failure	Woon et al. [70]
Lung	Populations of CD56^dim^CD16^+^ NK cells with different receptors could be found in the lungs of patients undergoing surgery for suspected lung cancer.	Brownlie et al. [71]
	Manipulated macrophage polarization by depletion of NK cells attenuates acute lung injury	Wu et al. [72]
	Depletion of NK cells or using NKG2D stress receptor blockade can alleviate acute lung injury.	Calabrese et al. [73]
Liver	Tim-3 enhances hepatocellular carcinoma (HCC) growth by blocking natural killer cell function.	Tan et al. [74]
	NASH have an increased expression of NKG2D on human NK cells	Stiglund et al. [75]
	NK cells killed of activated HSCs to anti-fibrotic	Melhem et al. [76]
	Natural killer cells reduce liver fibrosis by killing activated stellate cells	Radaeva et al. [77]

**Table 3 biomolecules-13-00748-t003:** The relationship between NK cells and kidney diseases, including ischemia–reperfusion injury, renal fibrosis, adriamycin nephropathy, aristolochic acid-induced nephropathy (AAN), renal cell carcinoma, and kidney transplantation.

Renal Disease Type	Function of NK Cells in Disease	Authors
Ischemia–reperfusion injury (IRI)	NK cells cause tubular epithelial cell (TEC) apoptosis and contribute to renal IRI after being activated by osteopontin.	Zhang et al. [106]
	NK cells can directly kill TECs in vitro, correlated with TEC expression of RAE-1 and NKG2D on NK cells.	Zhang et al. [97]
	Increased infiltration of NK1.1^+^ and CD4^+^NK1.1^+^ cells were observed 3 and 24 h after renal IRI.	Ascon et al. [98]
	TECs produced a high CXCR2 level that promoted neutrophil chemotaxis after binding CD137 from NK cells with CD137 ligand (CD137L) on TECs, and toll-like receptor (TLR) 2 ligands released from ischemic TECs induce NK cell recruitment.	Kim et al. [99]
	Depleting NK cells protects mice from kidney dysfunction during IRI, and tissue-resident NK cells promote AKI	Victorino et al. [100]
	TLR4 signaling affects the expression of the NKG2D ligands RAE-1 and MULT-1 on kidney cells to participate in the pathogenesis of renal IRI	Chen et al. [101]
	NK cells are essential in recruiting neutrophils during kidney IRI	Kim et al. [102]
Renal fibrosis	CD56^bright^ natural killer cells produce Interferon-γ (IFNγ), contributing to renal fibrosis and chronic kidney disease (CKD) progression	Law et al. [103]
Adriamycin nephropathy	NKs activating receptor NKG2D and its ligand RAE-1 are upregulated by AN	Zheng et al. [104]
Aristolochic acid-induced nephropathy (AAN)	Tissue-resident NK cells exacerbated tubulointerstitial fibrosis by activating transglutaminase 2 and syndecan-4 in the AAN model	Wee et al. [105]
Renal cell carcinoma	NK cells were found to express various inhibitory receptors (IRs) such as CD94/NKG2A receptor complex	Schleypen et al. [107]
	The function of NK cells can be predicted by NK cell infiltration level and the expression of markers (CD16 and cytotoxins) in renal cell carcinoma	Schleypen et al. [108]
Kidney Transplantation	NK cells can activate monocytes by secreting cytokines or directly cause renal microvascular inflammation, renal interstitial inflammation, renal tubular inflammation, etc.	Jasper Callemeyn et al. [20]

## Data Availability

Not applicable.

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
