# Peer review of "Natural Killer Cells, as the Rising Point in Tissues, Are Forgotten in the Kidney"

_biomolecules, 2023, doi:10.3390/biom13050748_

Round 1

Reviewer 1 Report

Dear Editor,

The following are my comments on manuscript ID biomolecules-2291239.

In this review, Ke Ma et al. summarized the subtypes and markers of natural killer (NK) cells and their roles in different organs of various diseases. The authors highlighted the potential functions of NK cells in renal pathophysiological conditions. By writing this review, the authors would like to attract the reader’s attention to NK cells in renal diseases. I think this is a well-organized review with comprehensive and the latest information. The following comments and correction of grammar errors may improve the quality of this review.

Figures:

1. In Figure 1, I appreciate the authors’ effort in making pretty illustrations showing the identification markers of different NK subtypes. However, the CD markers are labeled at various angles to match the receptors in cartoons, making it difficult for readers to catch the information. It may play the opposite effort in the presentation. I would suggest authors list all the markers next to or beneath the NK cells in the cartoon. Here is an example (Figure 1 of Natural Killer Cells and Their Role in Immunity, https://www.emjreviews.com/allergy-immunology/article/natural-killer-cells-and-their-role-in-immunity/).

2. In Figure 4, please list all the abbreviations in the figure legend.

3. In Figure 5, is the figure legend double-spaced? Please keep it the same way as other figure legends.

Grammar errors:

1. Line 37: “indicates” instead of “indicated”.

2. Line 71: “contributing to halting kidney disease progression and trying to” instead of “contributing to halt kidney disease progression and try to”.

3. Line 78: “B cells and T cells” instead of “B cells and T cell”.

4. Line 88: “humans and mice” instead of “humans and mic”.

5. Line 205: By contrast should lead a new sentence.

6. Line 220: “which connect” instead of “which connected”.

7. Line 245: “was” instead of “were”.

8. Line 270: “was” instead of “were”.

9. Please ensure all the markers are in proper and consistent fonts. For instance, “dim” should be superscript in “CD56dim”.

Author Response

Reviewer 1:

1.“In Figure 1, I appreciate the authors’ effort in making pretty illustrations showing the identification markers of different NK subtypes. However, the CD markers are labeled at various angles to match the receptors in cartoons, making it difficult for readers to catch the information. It may play the opposite effort in the presentation. I would suggest authors list all the markers next to or beneath the NK cells in the cartoon. Here is an example (Figure 1 of Natural Killer Cells and Their Role in Immunity, https://www.emjreviews.com/allergy-immunology/article/natural-killer-cells-and-their-role-in-immunity/)”

Reply: Thanks for the suggestion. We have listed all the markers next to the NK cells to give a better understanding of readers in Figure 1.

2.“In Figure 4, please list all the abbreviations in the figure legend.”

Reply: Thanks for your comments. We already add the abbreviations in the manuscript lines 305-308.

3.“In Figure 5, is the figure legend double-spaced? Please keep it the same way as other figure legends.”

Reply: Thanks for your comments. We are sincerely sorry for the carelessness. And we already modified the mistake.

4.“Line 37: “indicates” instead of “indicated”

Reply: Thanks for your comments. We are sincerely sorry for the carelessness. And we already modified the mistake.

5. “Line 71: “contributing to halting kidney disease progression and trying to” instead of “contributing to halt kidney disease progression and try to”

Reply: Thanks for your comments. We are sincerely sorry for the carelessness. And we already modified the mistake.

6. “Line 78: “B cells and T cells” instead of “B cells and T cell”

Reply: Thanks for your comments. We are sincerely sorry for the carelessness. And we already modified the mistake.

7. “Line 88: “humans and mice” instead of “humans and mic”

Reply: Thanks for your comments. We are sincerely sorry for the carelessness. And we already modified the mistake.

8. “Line 205: By contrast should lead a new sentence.”

Reply: Thanks for your comments. We are sincerely sorry for the carelessness. And we already modified the mistake.

9. “Line 220: “which connect” instead of “which connected”.”

Reply: Thanks for your comments. We are sincerely sorry for the carelessness. And we already modified the mistake.

10. “Line 245: “was” instead of “were”

Reply: Thanks for your comments. We are sincerely sorry for the carelessness. And we already modified the mistake.

11. “Line 270: “was” instead of “were”

Reply: Thanks for your comments. We are sincerely sorry for the carelessness. And we already modified the mistake.

12. “Please ensure all the markers are in proper and consistent fonts. For instance, “dim” should be superscript in “CD56dim

Reply: Thanks for your comments. We are sincerely sorry for the carelessness. And we already modified the mistake.

Reviewer 2 Report

This review article introduces several main classes of Natural killer (NK) cells which may involve in kidney diseases. This article brings basic and novel knowledge of NK cells to readers. However, the part that described the pathological role of NK cells in kidney diseases is not comprehensive. There are some suggestions.

1. The article title could be revised, especially “Forgotten in the Kidney”. In recent 5 years, there are more than 300 articles described or investigated the involvement of NK cells in kidney diseases. The title sounds focusing on more deeply review of the role of NK cells in kidney diseases, but the text content doesn’t look like that.         

2. Line 244-253, reference should be cited. If results of figure 3 are unpublished results provided by the authors, how to prove the findings are true? More experimental details should be provided in the legend of figure 3.  

Author Response

Dear Editor,

Many thanks for the information concerning our manuscript submitted to Biomolecules. The manuscript has been revised according to the comments from the reviewers, and the details are as follows:

Reviewer 2:

1. “The article title could be revised, especially “Forgotten in the Kidney”. In recent 5 years, there are more than 300 articles describing or investigating the involvement of NK cells in kidney diseases. The title sounds focusing on a more deep review of the role of NK cells in kidney diseases, but the text content doesn’t look like that.”

Reply: Thanks for your suggestion. In the beginning, we aim to illustrate the feature function of NK cells in the kidney compared with other organs. We already modified the title to “The current advances of natural killer cells, as a rising point in tissues, especially in the kidney” according to your suggestion.

2.“Line 244-253, reference should be cited. If the results of Figure 3 are unpublished and provided by the authors, how can the findings be true? More experimental details should be provided in the legend of Figure 3.”

Reply: Thank you for your comments. Figure 3 are unpublished results provided by us. We can provide primary data of flow cytometry. In addition, we added the legend to demonstrate the procedure in Figure 3.

Reviewer 3 Report

This is a very interesting review article. 

Authors have covered roles of NK cells in different tissues and disease conditions. 

Recently, the role of NK cells was assessed by many workers in COVID-19. 

Also, the NK cell-based therapies are not discussed.

This could be very interesting to see translational aspects of this information

Author Response

Dear Editor,

Many thanks for the information concerning our manuscript submitted to Biomolecules. The manuscript has been revised according to the comments from the reviewers, and the details are as follows:

Reviewer 3:

“This is a very interesting review article. Authors have covered roles of NK cells in different tissues and disease conditions. Recently, the role of NK cells was assessed by many workers in COVID-19. Also, the NK cell-based therapies are not discussed.

Reply: Thank you for your comments. We had already explored the function of NK cells in the renal inflammatory storm in our experiments. The next article will demonstrate the function of NK cells in inflammatory storms similar to COVID-19. At that time, we will show how CAR-NK affects kidney injury in inflammatory storms.

Reviewer 4 Report

The review by Ke Ma et al. is devoted to the interesting problem of the role of NK cells in kidney pathology.  The review as a whole makes a favorable impression, but a number of criticisms should be addressed before the review is accepted for publication.

1.       The role of NK cells in the kidneys is rather well studied. Quite a lot of reviews have been written covering certain problems of NK cells in the kidneys (for review see: https://pubmed.ncbi.nlm.nih.gov/34446934/, https://pubmed.ncbi.nlm.nih.gov/30972076/, https://pubmed.ncbi.nlm.nih.gov/34915041/, https://pubmed.ncbi.nlm.nih.gov/33387602/) What is the uniqueness and zest of this review? I would recommend authors focusing the review on a certain scientific problem.

2.       About 30% of the review is devoted to the presentation of the basic biology of NK cells. In my opinion, in 2023 this is not necessary.

3.       The list of references seems to be outdated. Currently, only 22% of cited articles have been published in the last 3 years. The reviews mentioned above (#1) should be analyzed and cited (with the exception of Turner et al., they are not included in the review).

4.       Fig 3, showing the results of flow cytometry in mice doesn't make any sense in a review on human NK cells

Author Response

Dear Editor,

Many thanks for the information concerning our manuscript submitted to Biomolecules. The manuscript has been revised according to the comments from the reviewers, and the details are as follows:

Reviewer 4:

  1. “The role of NK cells in the kidneys is rather well studied. Quite a lot o reviews have been written covering certain problems of NK cells in the kidneys (for review see: https://pubmed.ncbi.nlm.nih.gov/34446934/, https://pubmed.ncbi.nlm.nih.gov/30972076/, https://pubmed.ncbi.nlm.nih.gov/34915041/, https://pubmed.ncbi.nlm.nih.gov/33387602/) What is the uniqueness and zest of this review? I would recommend authors focusing the review on a certain scientific problem.”

Reply: Thank you for your comments. We have learned lots of knowledge about NK cells function during writing the manuscript and reviewer’ comments. Previous researches often demonstrated the function of NK cells in acute kidney disease or chronic kidney disease. In the review, we showed the pathogenesis of NK cells in more kidney diseases in renal cell carcinoma, doxorubicin nephropathy, renal fibrosis, chronic kidney disease, acute kidney injury, etc. Moreover, we list the function of NK cells in other organs to provide a novel thought into NK cells’ role comparing that in renal with that in the liver, adipose, lung, and uterus. We hope to the analogy of NK cells from other diseases to kidney diseases, thus accelerating the targeted treatment of NK cells in kidney diseases.

  1. “About 30% of the review is devoted to the presentation of the basic biology of NK cells. In my opinion, in 2023 this is not necessary.”

Reply: Thank you for your comments. In 2023, we have a comprehensive understanding of the basic physiological knowledge of NK cells. However, for non-NK cell researchers and clinical doctors, the understanding of NK cells may be still limited. We hope to provide innovative research ideas on NK cells for professional researchers as well as doctors through this article. 

  1. “The list of references seems to be outdated. Currently, only 22% of cited articles have been published in the last 3 years. The reviews mentioned above (#1) should be analyzed and cited (with the exception of Turner et al., they are not included in the review).”

Reply: Thanks for your suggestion. We have added the papers mentioned above (#1) and novel ones to update the list of references in lines 263 and 46.

  1. “Fig 3, showing the results of flow cytometry in mice doesn't make any sense in a review on human NK cells

Reply: Thanks for your suggestion. Due to the partial similarity of NK cell markers in mouse and human kidneys. And in basic research, the experiment of NK cells in mice is an advance in human exploration. So we hope to retain Figure 3, in which we used flow cytometry to explore the number of NK cells in mouse kidneys.

Round 2

Reviewer 4 Report

The authors have revised the article in accordance with the comments and in general it can be published in the journal. As for my minor remarks - I remain convinced that Fig. 3, which is a particular example of mouse NK cell flow cytometry, adds absolutely nothing to the review.